# Alcohol Use Disorders among Slovak and Czech University Students: A Closer Look at Tobacco Use, Cannabis Use and Socio-Demographic Characteristics

**DOI:** 10.3390/ijerph182111565

**Published:** 2021-11-03

**Authors:** Beata Gavurova, Viera Ivankova, Martin Rigelsky

**Affiliations:** 1Center for Applied Economic Research, Faculty of Management and Economics, Tomas Bata University in Zlín, Mostní 5139, 760 00 Zlín, Czech Republic; 2Faculty of Mining, Ecology, Process Control and Geotechnologies, Technical University of Košice, Letná 9, 042 00 Košice, Slovakia; viera.ivankova@tuke.sk; 3Faculty of Management, University of Prešov in Prešov, Konštantínova 16, 080 01 Prešov, Slovakia; martin.rigelsky@unipo.sk

**Keywords:** alcohol dependence, tobacco, cannabis, marijuana, smoking, COVID-19 pandemic, young adults, substance use, socio-demographic

## Abstract

The main objective of the research was to examine the associations between problematic alcohol use, tobacco use and cannabis use among Czech and Slovak university students during the early COVID-19 pandemic. The research sample consisted of 1422 participants from the Czech Republic (CZ) and 1677 from the Slovak Republic (SK). The analyses included university students who drank alcohol in the past year (CZ: 1323 (93%); SK: 1526 (91%)). Regarding the analysed measures, the Alcohol Use Disorders Identification Test (AUDIT) and its subscales, the Glover-Nilsson Smoking Behavioral Questionnaire (GN-SBQ) and the Cannabis Abuse Screening Test (CAST) were selected to identify substance-related behaviour. Age, gender and residence were included in the analyses as socio-demographic variables. Correlation and regression analyses were used to achieve the main objective of the research. The main results revealed that the use of tobacco and cannabis were positively associated with alcohol use disorders among Czech and Slovak university students. Additionally, males were more likely to report alcohol use disorders. In the Czech Republic, it was found that students living in dormitories were characterized by a lower AUDIT score. The opposite situation was found in the Slovak Republic. Czech and Slovak policy-makers are encouraged to develop alcohol use prevention programs for university students in line with these findings.

## 1. Introduction

Young people, especially university students, are a population group at risk of unhealthy and harmful behaviours in terms of the use of addictive substances such as alcohol, tobacco or cannabis [1,2,3]. The vulnerability of this population group is evidenced by a considerable prevalence of dependence to alcohol, tobacco and illicit drugs [4]. All these facts are the result of a new stage of life focused more on their personality, as they are curious and want to fit into the team, experience sensation and build their own social identity [5,6].

The combined use of addictive substances in the university environment is not an exceptional phenomenon either [7]. According to Nasui et al. [8], both male and female university drinkers engaged in other risky behaviours correlated with drinking. These patterns of behaviour have many consequences, whether it is a threat to health and life [9], reduced academic performance, missed classes and lower grades, memory blackouts, changes in brain function, lingering cognitive deficits, sexual assaults [10], but also poor mental health [11] or social problems [12]. Regarding determinants, it is well known that the male gender characteristic is a significant factor associated with increased alcohol use [13,14] and is therefore considered a predictor of alcohol use disorders [15,16]. This justifies respecting gender differences in research. In addition to gender characteristics, there are many other possible determinants of problematic alcohol use such as living away from parents’ home during semesters [1,17], parent attachment [18], smoking [15], mental health problems and satisfaction with life [19,20], or age of alcohol consumption onset [21]. The need to constantly monitor this problem is underlined by the fact that university students who suffer from problematic alcohol use with the risk of dependence are also characterized by the use of other addictive substances, such as tobacco, cannabis or cocaine [22]. All these factors can contribute to higher levels of alcohol use, which is an undesirable phenomenon in society. Thus, factors such as residence, age, gender, living conditions, smoking, or illicit substance use should be included in research into alcohol use disorders among university students.

It is true that the COVID-19 pandemic is an unknown situation, and more pressure can be expected in students’ lives. Young adults face various impulses of risky behaviour during the pandemic [23,24,25]. Thus, in the context of the pandemic, increased attention should be paid to the psychological distress that is associated with heavy drinking and a high-risk level of drug use among university students [26]. Jackson et al. [27] examined COVID-19-related changes in drinking among university student drinkers that were attributable to changes in context, particularly a shift away from heavy drinking with peers to lighter drinking with family. Their results revealed that reduced social opportunities and/or settings, limited access to alcohol, and reasons related to health and self-discipline were reflected in decreased alcohol use. On the other hand, increased alcohol use was attributed to greater opportunity (more time) and boredom and, to a lesser extent, to a lower perceived risk of harm and to cope with distress. As a result, poor mental health, as well as alcohol abuse, can be observed among university students [28]. All these facts indicate that substance use behaviour should be monitored, especially in a critical situation such as the COVID-19 pandemic.

All the above-mentioned findings indicate that alcohol use affects many dimensions in university students’ lives and is therefore a serious burden. In this regard, increased research attention should be paid to each region. Jia et al. [29] emphasized the need to investigate drug use among students also in terms of geographical differences, which may have strong links to socio-economic and demographic characteristics of the regions. This allows the design of successful interventions tailored to a geographical region with unique characteristics. As for the Czech Republic and the Slovak Republic, these countries share not only a common Central European space, but also a common history, culture, priorities, values and interests to strengthen the stability of society. This fact can also be applied in the field of addictology [30]. The Czech Republic and the Slovak Republic formed one unit, but were divided. On the other hand, each country behaves as an individual living organism, the core of which is a reflection of their own social needs and principles. For these reasons, their examination is warranted [31]. A previous study suggested potential differences in alcohol-related problems between individual regions of the countries of the former Czechoslovakia [32]. However, it was Nuevo et al. [33] who emphasized considerable differences in alcohol use between these two countries, which were politically gathered in the recent past. The Slovak Republic dominated in heavy drinkers. In this context, the abuse of addictive substances, including alcohol, was also considered a more relevant problem in the Slovak Republic compared to the Czech Republic in a sample of the general population [34]. In the pre-pandemic period, significant differences were also confirmed among university students from four countries, including the Czech Republic and the Slovak Republic [35].

Despite the fact that problematic alcohol use among students and its determinants is a well-researched issue in the world, the Czech Republic and the Slovak Republic are countries that have long overlooked and neglected this problem. Insufficiency can be seen not only in the research area, but also at the level of management of health policies targeted at a specific group of the population such as students, resulting in insufficient evidence-based interventions in the university environment. In other words, the application of research findings in practice is minimal in this region. Understanding the situation is particularly important in the current pandemic period for the development of successful strategies and programs aimed at reducing alcohol use among students, who are seen as the driving force of the economy in the future, but also as potential consumers of social support and health care. These facts reinforce the importance of the findings offered by the presented study, especially in the geographical region of the Czech Republic and the Slovak Republic. Without substantiated evidence, it is not possible to design and implement student-targeted programs that are lacking in the countries.

On this basis, the objective of the presented research was to examine the associations between problematic alcohol use, tobacco use and cannabis use among Czech and Slovak university students during the early COVID-19 pandemic. The study provides a valuable platform for information on problematic alcohol use accompanied by tobacco use, cannabis use and other characteristics of individuals, facilitating public health leaders’ decision-making.

## 2. Materials and Methods

### 2.1. Research Questions

This study provides a deeper insight into the issue of alcohol use disorders among university students and brings evidence from the Czech Republic and the Slovak Republic during the early COVID-19 pandemic. In the research, emphasis was placed on tobacco use, cannabis use and several socio-demographic characteristics. Based on the main aim, three research questions were formulated:

RQ 1: What are the differences in alcohol use disorders between Slovak and Czech university students?

RQ 2: What is the comorbidity between tobacco use, cannabis use, and alcohol use disorders among Czech and Slovak university students?

RQ 3: What are the associations of alcohol use disorders with tobacco use, cannabis use, age, male gender, and residency?

### 2.2. Research Sample and Data Collection Process

The participants involved in the presented research were Czech and Slovak university students. The research sample was formed on the basis of quota sampling with a focus on all Czech and Slovak universities. The purpose was to cover all universities, as well as all fields of study, with at least 30 observations. Prior to data collection, universities in both countries were mapped, including their number and their approximate size according to the number of students and fields of study. The ambition was to include every field of study, which succeeded. In this way, data was collected from 80% of universities. This fact makes the presented research unique, as such a large sample has not yet been studied in the examined region.

The data collection was performed using an online questionnaire distributed during the first wave of COVID-19 in 2020. The questionnaire was distributed through university representatives (rectors, vice-rectors, deans, vice-deans, university teachers and lecturers) and administrative staff who were asked to share it with students. At the same time, delegates of the student council for higher education were contacted with a request to distribute the questionnaire among students. Finally, students were addressed directly on social networks in student groups. The whole survey was conducted in both languages; thus, the participants were provided with the questionnaire in Czech and Slovak.

After removing irrelevant responses, the sample consisted of 1422 participants from the Czech Republic and 1677 from the Slovak Republic, with the data collection taking place during the first wave of the COVID-19 pandemic in 2020. When excluding responses, the criteria included disagreement with participation in the research, incorrect answer in the control item (one million has six zeros, while a numerical expression was also given), other than Slovak or Czech nationality, or study in another country. Table 1 shows the frequency of selected identifiers in the research sample.

As can be seen from Table 1, the research sample from the Slovak Republic was slightly more balanced than the research sample from the Czech Republic. Despite this fact, the total research sample could be considered sufficiently reliable for analytical processes. The identifiers included in the regression analysis were converted to the dichotomous scale described in the given part of the analytical process. Table 1 indicates that the most frequent form of study was the full-time form of study, which is a common form in the university environment in the examined region. Additionally, the research sample consisted mainly of participants studying a bachelor’s degree, which could be due to the fact that not every field of study continues with a master’s/engineering degree. Female students dominated over male students, and the most common residence type was living at home. Dormitory living was more frequent in the Slovak sample than in the Czech one. A considerable proportion of the participants lived in the countryside and in cities with 10,001 to 100,000 inhabitants.

### 2.3. Governance and Ethics

The research was approved by the ethics committee of the General University Hospital in Prague as individual research (Ref. 915/20 S–IV). At the beginning of the questionnaire, all important information on research and processing of personal data was provided. The survey was completely anonymous and personal data was protected. All participants involved in this research confirmed informed consent in the questionnaire. All aspects in this research were conducted with respect to the seventh revision of the World Medical Association Declaration of Helsinki [36] and the second revision of the Farmington Consensus [37].

### 2.4. Research Instruments and Variables

The analyses used in this research included variables identifying a participant’s attitude to substance use at a particular time. In this way, the analyses included a variable determining whether participants drank alcohol in the past year (CZ: No = 84 (5.9%), Yes = 1323 (93%), blank = 15 (1.1%); SK: No = 151 (9%), Yes = 1526 (91%)). Accordingly, the research sample consisted of 1323 Czech participants and 1526 Slovak participants. Subsequently, this research sample was the basis for a variable determining whether participants smoked tobacco in the last three months (CZ: No = 873 (66%), Yes = 450 (34%); SK: No = 1102 (72.2%), Yes = 424 (27.8%)) and for a variable determining whether they smoked cannabis in the past year (CZ: No = 1047 (79.1%), Yes = 276 (20.9%); SK: No = 1226 (80.3%), Yes = 300 (19.7%)).

The analyses included three indicators of substance use, namely alcohol, tobacco and cannabis. The Alcohol Use Disorders Identification Test (AUDIT) [38] was used to identify alcohol use. The AUDIT measure was developed to detect problematic alcohol use and its intensity. This brief tool consists of 3 domains (hazardous alcohol use, alcohol use with dependence symptoms, harmful alcohol use) and of 10 items. The AUDIT items are scored from 0 to 4, and the total score is the sum of the individual items. The higher the total score, the higher the level of risk of alcohol use disorder. The risk levels are identified as follows (Zone—recommended intervention): (i) low risk without potential alcohol use disorder (Zone I—alcohol education), (ii) mild risk (Zone II—simple advice), (iii) moderate risk (Zone III—simple advice plus brief counselling and continued monitoring), and (iv) moderate/severe risk (zone IV—referral to specialist for diagnostic evaluation and treatment). The AUDIT measure is commonly used in the professional and scientific community. This is evidenced by several studies, in which this tool was used also in a sample of university students [39,40]. In the Slovak Republic, its reliability was verified by Janovská et al. [41].

The Glover-Nilsson Smoking Behavioral Questionnaire (GN-SBQ) [42] was used to identify tobacco use. This simple 11-item questionnaire was able to assess behavioural dependence; in other words, to identify aspects of smoking dependence that are behavioural in nature. The GN-SBQ measure is commonly used by physicians, health care providers, and tobacco interventionists. The following answers were provided to GN-SBQ items: (0) not at all, (1) somewhat, (2) moderately so, (3) very much so, and (4) extremely so. This measure provides a total score, and the higher the total score, the higher the dependence. Based on the total score, dependence is identified at the following levels: mild (<12), moderate (12–22), strong (23–33), and very strong (>33). Thus, high scores in the GN-SBQ measure indicate the need for greater emphasis by physicians on behavioural management.

The Cannabis Abuse Screening Test (CAST) [43,44] was used to identify cannabis use. The CAST measure provides psychometric properties for assessing problematic forms of cannabis use among young people and for identifying patterns of cannabis use leading to negative social or health consequences for individuals. This short measure consists of six items. The CAST items offered the following answers: (0) never, (1) rarely, (2) from time to time, (3) fairly often, (4) very often.

### 2.5. Statistical Analysis

The following statistical methods were used to achieve the main objective of this research. Descriptive analysis was used to present the basic statistical characteristics of alcohol-related variables (mean, median, variance (Var), standard deviation (St. Dev.), interquartile range (IQR), minimum (Min), maximum (Max)). The significance of differences between countries with respect to data characteristics was investigated using the nonparametric Mann–Whitney U test. Due to the nature of the data, Spearman’s correlation coefficient (ρ) was used to assess the relationships. Two regression models were used to evaluate the effects, namely the ordinary least squares (OLS) model and the negative binomial generalized linear model (NB) model [45]. Significant heteroscedasticity occurred in the OLS models; therefore, a robust estimation based on the HC3 estimator was preferred.

Analytical calculations were performed using the programming language R version 4.0.2 (RStudio, Inc., Boston, MA, USA), nickname: Taking off Again [46].

## 3. Results

This section presents the results of a descriptive analysis in order to offer a closer look at selected variables and point out a current situation in the examined issue. This section is also devoted to the examination of the relationships between alcohol use disorders and tobacco use, cannabis use and selected socio-demographic characteristics of the participants. This made it possible to map the situation in the region, where this problem has been overlooked for a long time and a deeper insight into the issue was lacking. In addition, the results help to point out the comorbidity and the association between substance use among Czech and Slovak university students during the early COVID-19 pandemic.

Table 2 shows the results of the descriptive analysis of the total AUDIT score and its individual subscales. With a focus on the statistical measures of the central tendency for *AUDIT Total, AUDIT Hazardous Alcohol Use* and *AUDIT Dependence Symptoms*, it was possible to observe the fact that Czech participants dominated over Slovak participants. Hazardous alcohol use and dependent alcohol use indicate a risk of mild and moderate/severe alcohol-related disorders. In terms of *AUDIT Harmful Alcohol Use*, the opposite situation was found, i.e., a higher mean score was identified for Slovak participants. Regarding the mean values of the total AUDIT score, it could be stated that Czech and Slovak students reported a low risk of alcohol use disorder during the early COVID-19 pandemic. However, it should be noted that the scores were almost on the threshold between low and mild risk.

The nonparametric Mann–Whitney test of differences was applied to these data, while significant differences were revealed only in *AUDIT Hazardous Alcohol Use* (statistic: 951,900, *p*-value: 0.008) and in *AUDIT Dependence Symptoms* (statistic: 944,764, *p*-value: <0.001).

Table 3 shows the descriptive analysis results of *GN-SBQ* and *CAST* measures, in general as well as for individual countries. The values of the central tendency measures were higher in the Slovak Republic, especially in the case of *GN-SBQ*. Based on the mean values of the total *GN-SBQ* score, a moderate smoking dependence in behavioural nature was found in both countries during the first wave of the COVID-19 pandemic. However, the moderate level of dependence ranged from 12 to 22, and therefore it could be stated that the Czech participants reported an almost threshold value between mild and moderate dependence. By focusing on the median of the total *CAST* scores, the measured values did not indicate a high risk of cannabis abuse among university students in both countries during the early COVID-19 pandemic.

Regarding the data on tobacco and cannabis use, the nonparametric Mann–Whitney test of differences did not show any significant differences between countries.

The following analytical procedures are devoted to the relationships between problematic alcohol use and tobacco and cannabis use. Thus, the analyses included the AUDIT measure and its subscales, but also the GN-SBQ measure and the CAST measure.

Table 4 shows the results of the correlation analysis between the investigated variables related to addictive substances. The first column (ρ) provides the rate of correlation, the second column (Sig.) shows a p-value and the third column (N) offers information on the number of observations. At this point, it should be noted that the analysis included data on participants who smoked tobacco in the last three months (*GN-SBQ*) and also drank alcohol in the past year, as well as data on participants who smoked cannabis (*CAST*) and also drank alcohol in the past year. In general, there were more significant correlations in the Slovak Republic than in the Czech Republic. The rate of correlations could be considered as low to medium.

The alcohol-related variables presented above were dependent variables in the regression analysis. Independent variables were represented by variables such as *Smoking* (tobacco smokers in the last three months = 1, tobacco non-smokers = 0), *Cannabis* (cannabis smokers in the past year = 1, cannabis non-smokers = 0), *Age* (CZ: mean = 24.7, median = 23, standard deviation = 6.08; SK: mean = 23.36, median = 22, standard deviation = 4.24), *Male* (males = 1, females = 0), *Countryside* (countryside = 1, city = 0), *Dormitory* (dormitory = 1, other than dormitory = 0).

In the following part of the analytical procedure, the results of regression models (the OLS model and the NB model) were presented for individual countries. Before using the regression models, the assumptions of the application of the models were first evaluated, predominantly for the OLS model. A multicollinearity was tested by the variance inflation factor (VIF) method, while the highest value was measured for *Cannabis* (VIF: CZ = 1.09; SK = 1.11). The constancy of variability of the residues was tested using the Breusch–Pagan test, and significant heteroscedasticity was found in all analysed cases. On this basis, the HC_3_ estimator was used to estimate the coefficients of the OLS model.

Table 5 presents the results of the OLS and NB regression models, confirming the significant associations of the AUDIT indicators with selected variables (*Smoking, Cannabis, Age, Male, Countryside, Dormitory*). In the analysed cases of *Smoking* and *Cannabis*, a significant positive association was found in all AUDIT indicators in the Czech Republic, as well as in the Slovak Republic. This finding points to the fact that the use of tobacco and cannabis was positively associated with problematic alcohol use among Czech and Slovak university students during the early COVID-19 pandemic. In terms of *Age*, significant negative associations could be observed. For Czech university students, *Age* was significantly and negatively associated with *AUDIT Total*, *AUDIT Hazardous Alcohol Use* and *AUDIT Harmful Alcohol Use*. In contrast, only one significant negative association was found in Slovak university students, namely between *Age* and *AUDIT Hazardous Alcohol Use*. *Male* showed a positive and significant coefficient in all of the analysed cases. Thus, a male gender was positively associated with problematic alcohol use in the Czech Republic, as well as in the Slovak Republic, during the first wave of the COVID-19 pandemic. In contrast, no effect at the significance level of α < 0.05 was found for *Countryside*. The only discrepancy between the examined countries in terms of direction of associations was observed in *Dormitory*. This variable showed a negative association with *AUDIT Total* and *AUDIT Harmful Alcohol Use* in the Czech Republic. On the other hand, a significant positive association with all AUDIT indicators was identified in the Slovak Republic.

## 4. Discussion

### 4.1. Problematic Alcohol Use in the Czech Republic and the Slovak Republic

Based on the main results of the difference analysis, significant differences in problematic alcohol use were found between the examined countries in domains such as *AUDIT Hazardous Alcohol Use* and *AUDIT Dependence Symptoms*. This finding clarifies the answer to the research question 1 (RQ1) and encourages further investigation after the pandemic. A higher value was measured in the Czech Republic. Regarding the pre-pandemic period, very similar results were identified by Kalina et al. [35], who examined data from 2016 and found that Czech university students reported the highest mean AUDIT scores in terms of hazardous alcohol use and alcohol use with dependence symptoms compared to university students from the Slovak Republic, Hungary and Lithuania. On the other hand, this finding contradicts the evidence revealed by Nuevo et al. [33] and Slachtová et al. [34], who considered the Slovak Republic to be a country with greater alcohol-related problems. The discrepancy may have been due to the fact that the presented research was aimed only at university students. In explaining the results of this study, it can be assumed that Czech and Slovak students may have had different restrictions and educational conditions during the first wave of the COVID-19 pandemic. This could lead to different drinking opportunities and behaviours among students in these two countries.

In general, the mean value of the total AUDIT score was 6.12 in the Czech Republic and 6.05 in the Slovak Republic, indicating a low level of risk (Zone I—alcohol education). Although this value does not represent a potential alcohol use disorder, it should be emphasized that the limit value between Zone I and Zone II with a mild risk of alcohol use disorder is 7. Thus, this unhealthy pattern should be monitored among Czech and Slovak university students. Tóthová [47] found similar results among Slovak university students in the pre-pandemic period, while most students were included in Zone I. The results of this study are consistent with the results of Kalina et al. [35], who measured very similar values in all individual AUDIT domains based on pre-pandemic data. This indicates that Czech and Slovak university students did not change their drinking during the pandemic. The explanation can be found in the fact that, despite bars, pubs, cafes, clubs and restaurants were closed, individuals could still drink alcohol from the store [48,49]. At the same time, off-premises alcohol use may not be apparent in the short time since the onset of the pandemic [49].

### 4.2. Tobacco and Cannabis Use in the Czech Republic and the Slovak Republic

When assessing tobacco use measured by the *GN-SBQ* score, the participants from both countries reported higher values (CZ: 12.21, SK: 14.17), which could be included in the second level of behavioural dependence, reflecting the prevalence of smoking in Central and Eastern Europe from the pre-pandemic period [50,51]. As this was a mean value, attention should be paid to this smoking-related indicator. Regardless of the pandemic period, a moderate smoking dependence found in both countries indicates the need for greater emphasis by physicians on behavioural management in the university environment. The *CAST* score did not indicate a high risk of cannabis abuse among university students; however, the importance of monitoring should also be emphasized for this pattern of unhealthy behaviour [52,53].

### 4.3. Associations of Alcohol Use Disorders with Tobacco Use, Cannabis Use and Selected Socio-Demographic Characteristics

The correlation analysis of the links between the total AUDIT score, including its subscales, and the tobacco (*GN-SBQ*) and cannabis (*CAST*) indicators revealed a significant positive correlation with a low to medium rate in most of the analysed cases. This finding can be interpreted as meaning that higher scores in tobacco and cannabis use may be associated with higher scores in problematic alcohol use. In the Slovak Republic, a significant correlation was more common than in the Czech Republic. This is in line with the finding of Lotrean et al. [54], who used the linear regression analyses and confirmed a positive correlation between e-cigarette use, smoking, experimentation with alcohol intoxication, and the use of illicit drugs among Romanian university students. Thus, it is possible to speak of a comorbidity of other adverse health behaviours, such as smoking and alcohol abuse in university students, as confirmed by Zadarko-Domaradzka et al. [55] in the Carpathian Euroregion. This finding clarifies the answer to the research question 2 (RQ2), and an explanation can be found in the well-known fact that one type of unhealthy behaviour encourages young people to engage in further unhealthy behaviour.

Based on the regression analysis examining the associations of selected indicators related to addictive substances and socio-demographic characteristics, it can be stated that tobacco use, cannabis use and male gender were positively associated with problematic alcohol use in Czech university students. An interesting finding in the Czech Republic was the fact that dormitory living was negatively associated with problematic alcohol use in terms of domains such as *AUDIT Total* and *AUDIT Harmful Alcohol Use*. Almost all results in the Slovak Republic can be interpreted in a similar way with one difference in dormitory living. Thus, Slovak university students living in dormitories showed more frequent alcohol use disorders, which is not in accordance with the results of Tóthová [47]. Many factors may have contributed to this discrepancy, and further research is needed to better understand the results. The fact is that living in dormitories offers more opportunities for risky behaviours, including drinking alcohol and using other addictive substances, than living at home with parents [1,17]. Nevertheless, the COVID-19 pandemic was an unknown situation and various measures were taken across the countries, which also affected the process of education at universities. One of the factors contributing to the discrepancy may have been different anti-coronavirus interventions in the university environment in the Czech Republic and the Slovak Republic.

In general, Dos Reis and de Oliveira [22] found similar results to those in this study and emphasized that Brazilian university students suffering from problematic alcohol use at risk of dependence also tended to use other addictive substances, such as tobacco, cannabis or cocaine. De Oliveira et al. [56] and El Ansari et al. [57] also revealed a significant association between illicit drug use and tobacco use, as well as episodes of binge drinking. An explanation can be found in the well-known fact that the use of one addictive substance can in itself be a risk factor for the use of other addictive substances, especially in young people. The reason for this situation in university students may be lower levels of self-regulation and higher levels of normative beliefs [35].

Based on the results in this study, it can be concluded that protective behavioural strategies and programs that are effective against alcohol-related harms should be improved by adopting gender differences and establishing gender-specific standards [58]. In this study, males were more likely to report alcohol use disorder than females. These findings may be the result of the lifestyle of male students and having less barriers than female students. A possible explanation may also be that males perceive alcohol-related problems less negatively than females [59]. In this regard, female students may be less exposed to problematic alcohol use. There is also evidence that alcohol-related protective factors predominate in females. They perceive greater social sanctions and are less characterized by factors of excessive drinking, such as aggressiveness, uncontrollable behaviour, sensation-seeking, and others [60]. Therefore, Czech and Slovak prevention programs should take this finding into account and place more emphasis on male students. By comparing the results with the results of studies from other countries, it is possible to observe consistency with the Polish study, in which smoking and male gender characteristics proved to be the main determinants of harmful and hazardous drinking [15]. At this point, it is possible to emphasize the similarity of the countries belonging to the Visegrad Group. Similar findings were also identified in studies conducted by Lemma et al. [13], Năsui et al. [14], or Benjet et al. [16].

It is also necessary to take into account the environment and related social contexts for drinking in which university students live. It is possible to agree with Mereu et al. [17], who confirmed that living away from parents’ home is associated with problematic alcohol use. However, this study showed a discrepancy between the examined countries in terms of the trajectory of the association between living in dormitories and problematic alcohol use. Especially for Slovak students living in dormitories, contextual factors may shape their expectations about the effects of alcohol at the daily level, which may be important to address in event-level interventions to reduce hazardous drinking in young adults [61]. During the COVID-19 pandemic, the perception of peers’ changes in alcohol-related behaviour is strongly linked to changes in students’ own alcohol use, and this fact needs to be taken into account when developing effective strategies and prevention programs [62]. This agrees with the results of Alves et al. [63], who found that current residence, drinking peers, alcohol-related knowledge and attitudes about alcohol have a statistically significant effect on the probability of developing a risky pattern of alcohol use among university students. This can also explain the discrepancy between results in the Czech Republic and the Slovak Republic. All these aspects form the university environment in which students find themselves. Living situation seems to play an important role in the problem [1,64], and the dormitories offer many elements that can affect drinking patterns among students, regardless of direction. All these findings provide an answer to the research question 3.

### 4.4. Implications for Areas of Public Health and Substance Use Disorders among Young Adults

The study contributes important findings on the shape of the associations between alcohol use disorders, tobacco use, cannabis use and selected socio-demographic characteristics in the student population to the knowledge of the current situation in the Czech Republic and the Slovak Republic. In both countries, this problem is largely overlooked and insufficiently addressed among young people. The authors therefore appeal to alcohol prevention among university students, with a closer focus on current users of tobacco and cannabis, as they appear to be most at risk of alcohol use disorders. It is also recommended that decision-makers and policy-makers develop alcohol use prevention programs that take into account male gender and residence.

Strategies aimed at young adults and the university environment should be an essential part of national substance use control policies. In this sense, it is possible to talk about education, the promotion of a healthy lifestyle, but also the provision of appropriate assistance in the form of counselling. The public measures and interventions to stop drinking without being exposed to stigma appears to be an effective step [65,66]. These interventions can also lead to an increase in the level of health literacy among students, which is an essential element of public health [67]. For instance, in terms of higher levels of health literacy, responsible alcohol consumption, awareness of alcohol-related risks and understanding health information are essential for improving health. Factors of living conditions, self-perceived health, and the frequency of alcohol use play an important role in the context of health literacy [68,69]. Universities and other educational institutions can be the perfect environment for raising students’ awareness of the problems of using not only alcohol, but also other addictive substances that are associated with alcohol use disorders (tobacco and cannabis). In respect to the continuing threat of the COVID-19 pandemic, it is essential that public health leaders, researchers, health professionals and parents understand the importance of improving health literacy with an emphasis on substance use, as they can have lasting effects on the health and social position of university students [9,10,11,12]. In addition to providing information on current research, risks, guidance on early identification and other interventions for alcohol consumers, university education should not be left without motivating students to lead a healthy lifestyle without drinking alcohol, but also without smoking tobacco and cannabis. Promoting sports activities and a healthy lifestyle as competitive activities compared to alcohol use seems to be the most appropriate strategy in this challenging period of student life.

The main challenges for Czech and Slovak leaders in the field of public health are the removal of barriers and the development of effective intervention strategies. Alcohol use disorder is a serious mental problem with consequences in the future, and public health leaders should seek to integrate young people’s mental health into all dimensions of social and health policies, strategies and interventions. With regard to mental health, efforts should therefore focus on implementing students’ alcohol-related problems into general health policy, improving public awareness of alcohol-related risks, and redistributing resources to high-priority needs and vulnerable groups in the university environment [70].

All of these recommendations can help prevent more serious problems such as the incidence of alcohol dependence across the population, but also support future economic drivers and reduce the potential health costs of alcohol.

## 5. Conclusions

The main objective of the presented research was to examine the associations between problematic alcohol use, tobacco use and cannabis use among Czech and Slovak university students during the early COVID-19 pandemic. This study answered three research questions. Following them, it was possible to confirm the differences in alcohol use disorders between Slovak and Czech university students, the comorbidity between tobacco use, cannabis use, and alcohol use disorders among Czech and Slovak university students, as well as the associations of alcohol use disorders with tobacco use, cannabis use, age, male gender, and residency. This study filled a research gap in the Czech Republic and the Slovak Republic, where increased attention is needed towards vulnerable young adults and their unhealthy patterns of behaviour. Efforts to solve this problem may translate into benefits in the future, otherwise a greater burden on society can be expected, as problematic alcohol use may result in alcohol dependence, which is a serious social problem. This study highlights the importance of alcohol prevention, the availability of information on the risks and consequences associated with alcohol use, and information on appropriate counselling centres for university students in the Czech Republic as well as in the Slovak Republic. The findings of this study encourage the implementation of effective and evidence-based strategies, which are more than necessary in these countries. A valuable platform of results revealed by the presented research can help this effort.

### Future Directions and Limitations

Mapping substance abuse among university students is important, but it is also very important to implement effective measures in practice and compare results. For this reason, future efforts should focus on examining the problem following the end of the COVID-19 pandemic, with an emphasis on established measures at universities. Additionally, more similar countries should be included in the research. For instance, the countries of the Visegrad Group offer opportunities to understand their culture, politics and other specifics.

The limitations of the research include the fact that data collection was performed during a distanced form of education, which could be different and with different demands on students across fields of study in the Czech Republic as well as the Slovak Republic. The difficulty could affect their unhealthy patterns. A possible limitation is also the disproportionate nature of the research sample. Thus, there was a higher proportion of females. However, this limitation need not be considered disruptive to the results and value of knowledge.

## Figures and Tables

**Table 1 ijerph-18-11565-t001:** Research sample identifiers.

Frequency	CZ		SK	
	N	%	N	%
Field of study:				
Education	277	19.5	80	4.8
Humanities and arts	101	7.1	78	4.7
Social, economic and legal sciences	665	46.8	671	40.0
Natural science	50	3.5	73	4.4
Design, technology, production and communications	93	6.5	164	9.8
Agricultural and veterinary sciences	67	4.7	53	3.2
Health service	54	3.8	180	10.7
Services (tourism, sports, security, transport, logistics, …)	69	4.9	240	14.3
Informatics, mathematics, information and communication technologies	46	3.2	138	8.2
Form of study:				
Full-time	1041	73.2	1550	92.4
Part-time	381	26.8	127	7.6
Degree of study:				
Bachelor’s	658	46.3	1140	68.0
Master’s/Engineering	380	26.7	428	25.5
Combined (Bachelor’s and Master’s/Engineering)	50	3.5	41	2.4
Doctoral	334	23.5	68	4.1
Gender:				
Male	349	24.5	606	36.1
Female	1073	75.5	1071	63.9
Residence—university:				
Dormitory	243	17.1	702	41.9
Private accommodation	287	20.2	139	8.3
With family	202	14.2	68	4.1
With a friend	40	2.8	30	1.8
At home	650	45.7	738	44.0
Residence—home:				
Countryside	457	32.1	823	49.1
City with up to 10,000 inhabitants	254	17.9	198	11.8
City of 10,001 to 100,000 inhabitants	459	32.3	525	31.3
City of 100,001 to 1,000,000 inhabitants	169	11.9	119	7.1
City with over 1,000,001 inhabitants	83	5.8	12	0.7

Note: N—number, CZ—Czech Republic, SK—Slovak Republic.

**Table 2 ijerph-18-11565-t002:** Descriptive statistics of selected alcohol-related variables.

	AUDIT Total			AUDIT Hazardous Alcohol Use			AUDIT Dependence Symptoms			AUDIT Harmful Alcohol Use		
	ALL	CZ	SK	ALL	CZ	SK	ALL	CZ	SK	ALL	CZ	SK
Mean	6.08	6.12	6.05	3.58	3.64	3.52	0.59	0.63	0.55	1.92	1.85	1.98
Median	5.00	5.00	5.00	3.00	4.00	3.00	0.00	0.00	0.00	1.00	1.00	1.00
Var	21.85	20.30	23.21	4.15	3.60	4.61	1.36	1.30	1.42	5.85	5.33	6.30
St. Dev.	4.67	4.51	4.82	2.04	1.90	2.15	1.17	1.14	1.19	2.42	2.31	2.51
IQR	5.00	5.00	5.00	3.00	3.00	3.00	1.00	1.00	1.00	3.00	3.00	3.00
Min	0.00	0.00	0.00	0.00	0.00	0.00	0.00	0.00	0.00	0.00	0.00	0.00
Max	30.00	28.00	30.00	12.00	11.00	12.00	10.00	8.00	10.00	16.00	13.00	16.00

Note: AUDIT—Alcohol Use Disorders Identification Test, CZ—Czech Republic, SK—Slovak Republic, Var—variance, St. Dev.—standard deviation, IQR—interquartile range, Min—minimum, Max—maximum.

**Table 3 ijerph-18-11565-t003:** Descriptive statistics of selected variables related to tobacco and cannabis.

	GN-SBQ			CAST		
	ALL	CZ	SK	ALL	CZ	SK
Mean	13.15	12.21	14.17	3.58	3.56	3.60
Median	13.00	12.00	14.50	2.00	2.00	2.00
Var	66.83	76.23	55.09	17.20	16.63	17.94
St. Dev.	8.17	8.7.	7.42	4.15	4.07	4.24
IQR	13.00	13.00	11.00	5.00	5.00	5.00
Min	0.00	0.00	0.00	0.00	0.00	0.00
Max	41.00	41.00	31.00	22.00	18.00	22.00

Note: GN-SBQ—Glover-Nilsson Smoking Behavioral Questionnaire, CAST—Cannabis Abuse Screening Test, CZ—Czech Republic, SK—Slovak Republic, Var—variance, St. Dev.—standard deviation, IQR—interquartile range, Min—minimum, Max—maximum.

**Table 4 ijerph-18-11565-t004:** Correlation analysis.

Spearman ρ	CZ			SK		
	ρ	Sig.	N	ρ	Sig.	N
The Glover-Nilsson Smoking Behavioral Questionnaire (GN-SBQ)						
AUDIT Total	0.098	0.038	450	0.242	<0.001	424
AUDIT Hazardous Alcohol Use	0.091	0.053	450	0.159	0.001	424
AUDIT Dependence Symptoms	0.070	0.137	450	0.179	<0.001	424
AUDIT Harmful Alcohol Use	0.100	0.035	450	0.229	<0.001	424
Cannabis Abuse Screening Test (CAST)						
AUDIT Total	0.143	0.017	276	0.243	<0.001	300
AUDIT Hazardous Alcohol Use	0.109	0.072	276	0.145	0.012	300
AUDIT Dependence Symptoms	0.075	0.214	276	0.176	0.002	300
AUDIT Harmful Alcohol Use	0.145	0.016	276	0.265	<0.001	300

Note: AUDIT—Alcohol Use Disorders Identification Test, CZ—Czech Republic, SK—Slovak Republic, ρ—correlation rate, Sig.—significance, N—number.

**Table 5 ijerph-18-11565-t005:** Regression analysis.

DV	AUDIT Total		AUDIT Hazardous Alcohol Use		AUDIT Dependence Symptoms		AUDIT Harmful Alcohol Use	
Model	OLS	NB	OLS	NB	OLS	NB	OLS	NB
**CZ**								
Intercept	5.43 ^†^	1.73 ^†^	3.37 ^†^	1.23 ^†^	0.37 ***	−0.88 ^†^	5.43 ^†^	0.59 ^†^
Smoking	2.47 ^†^	0.39 ^†^	1.01 ^†^	0.26 ^†^	0.54 ^†^	0.82 ^†^	2.47 ^†^	0.49 ^†^
Cannabis	2.44 ^†^	0.34 ^†^	0.79 ^†^	0.19 ^†^	0.47 ^†^	0.59 ^†^	2.44 ^†^	0.52 ^†^
Age	−0.04 **	−0.01 ***	−0.02 ***	−0.01 **	<0.001	−0.01	−0.04 **	−0.02 **
Male	1.62 ^†^	0.24 ^†^	1.04 ^†^	0.26 ^†^	0.16 **	0.26 **	1.62 ^†^	0.22 ***
Countryside	0.38	0.05	0.1	0.02	0.04	0.08	0.38	0.11
Dormitory	−0.7 ***	−0.12 **	−0.19	−0.05	−0.1	−0.22	−0.7 ***	−0.23 **
R^2^	0.197	−	0.201	−	0.11	−	0.124	−
R^2^ Adjusted	0.194	−	0.198	−	0.106	−	0.12	−
Nagelkerke	−	0.2	−	0.178	−	0.112	−	0.11
AIC	−	6974.4	−	5114.9	−	2733	−	4747.5
**SK**								
Intercept	4.06 ^†^	1.5 ^†^	2.88 ^†^	1.09 ^†^	−0.08	−1.57 ^†^	4.06 ^†^	0.33
Smoking	3.12 ^†^	0.48 ^†^	1.31 ^†^	0.34 ^†^	0.53 ^†^	0.83 ^†^	3.12 ^†^	0.6 ^†^
Cannabis	3.01 ^†^	0.4 ^†^	1.09 ^†^	0.26 ^†^	0.45 ^†^	0.61 ^†^	3.01 ^†^	0.59 ^†^
Age	−0.02	−0.01	−0.02 **	−0.01 **	0.01	0.01	−0.02	−0.01
Male	1.82 ^†^	0.3 ^†^	1.05 ^†^	0.29 ^†^	0.28 ^†^	0.44 ^†^	1.82 ^†^	0.26 ^†^
Countryside	−0.11	−0.04	0.06	0.01	−0.02	−0.12	−0.11	−0.11 *
Dormitory	1.09 ^†^	0.19 ^†^	0.41 ^†^	0.11 ^†^	0.21 ^†^	0.42 ^†^	1.09 ^†^	0.28 ^†^
R^2^	0.246	−	0.228	−	0.104	−	0.162	−
R^2^ Adjusted	0.243	−	0.225	−	0.1	−	0.158	−
Nagelkerke	−	0.254	−	0.226	−	0.108	−	0.141
AIC	−	8036.8	−	6092.3	−	2822.5	−	5570.3

Note: *—*p*-value < 0.1; **—*p*-value < 0.05; ***—*p*-value < 0.01; ^†^—*p*-value < 0.001.

## Data Availability

Not applicable.

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
