# Peer review of "Alcohol Use Disorders among Slovak and Czech University Students: A Closer Look at Tobacco Use, Cannabis Use and Socio-Demographic Characteristics"

_ijerph, 2021, doi:10.3390/ijerph182111565_

Round 1
Reviewer 1 Report
This study goal is to predict whether tobacco use and cannabis use precede alcohol use dependency among students in the Czech Republic. Given the cross-sectional study design, this aim cannot be addressed. Given the fact that there are only scarce papers on alcohol dependency in this region, I suggest phrasing a different research question: 1. What are the comorbidity between tobacco use, cannabis use, and alcohol dependency among students in higher education in this region? 2. What are the differences in the prevalence of alcohol dependency between Slovak and Czech (and provide literature on these potential differences); 3. What are the associations between type of residency (dormitories or home) and cigarettes, cannabis, and alcohol use while gender is controlled?
The COVID-19 context is mentioned mainly in the introduction, but it creates a distraction, as the authors didn’t examine any variable related to COVID such as level of exposure to the virus, anxiety, loneliness, economic constraints, none.
There are also concerns related to the lack of conceptual rationale for the comorbidity, for the differences between the two groups/countries, and the inappropriate use of causality language. The description of the data collection and the sampling methods is vague, the order of the results is unusual- begins with prediction instead of descriptive findings, and the discussion provides a description without any explanations of the findings.
Author Response
Dear Reviewer,
First of all, we would like to thank you for the time spent evaluating the manuscript. We appreciate your evaluation and we consider it to be a significant motivation for our further work. We have carefully considered your comments and incorporated them into the revised manuscript hoping that the changes are in line with your expectations. The responses to your comments and recommendations can be found in the following text and all changes in the revised manuscript are highlighted in yellow.
Your review has been inspiring, and we see that this has greatly improved the quality of our study. We would like to thank you for the constructive feedback and for the opportunity to revise our manuscript.
Kind regards,
Beata Gavurova
Comments and Suggestions for Authors
This study goal is to predict whether tobacco use and cannabis use precede alcohol use dependency among students in the Czech Republic. Given the cross-sectional study design, this aim cannot be addressed. Given the fact that there are only scarce papers on alcohol dependency in this region, I suggest phrasing a different research question: 1. What are the comorbidity between tobacco use, cannabis use, and alcohol dependency among students in higher education in this region? 2. What are the differences in the prevalence of alcohol dependency between Slovak and Czech (and provide literature on these potential differences); 3. What are the associations between type of residency (dormitories or home) and cigarettes, cannabis, and alcohol use while gender is controlled?
- Response: Thank you for your study summary and for your willingness to help us. Also, thank you for these valuable recommendations that have been accepted and included in the revised manuscript. We fully agree with your view on the study. In the revised manuscript, the research questions have been modified and the study as a whole have been adapted to them. The Introduction section also contains literature on differences in alcohol use between the Czech Republic and the Slovak Republic.
The COVID-19 context is mentioned mainly in the introduction, but it creates a distraction, as the authors didn’t examine any variable related to COVID such as level of exposure to the virus, anxiety, loneliness, economic constraints, none.
- Response: Thank you for your comment, which has been taken into account when revising the study. The Introduction as a whole has been modified and we have tried to present COVID-19 only as a period that needs to be examined and in which the presented research took place. Subsequently, the fact that the results belong to the COVID-19 period has been emphasized throughout the study. Finally, in the Discussion section, a comparison of the results between the pre-pandemic period and the pandemic period has been provided. This made it possible to capture COVID-19 in terms of time and not its impact on society.
There are also concerns related to the lack of conceptual rationale for the comorbidity, for the differences between the two groups/countries, and the inappropriate use of causality language. The description of the data collection and the sampling methods is vague, the order of the results is unusual- begins with prediction instead of descriptive findings, and the discussion provides a description without any explanations of the findings.
- Response: Thank you for your comments. All of them have been incorporated into the revised manuscript. The conceptual rationale for the research has been emphasized throughout the text in the Introduction section, and the rationale for examining the differences between these two countries has been emphasized at the end of the Introduction section. Also, the language of causality has been corrected as recommended. In the revised manuscript, the interpretations are formulated in the light of associations. The Methodology section has been greatly improved. Thus, the description of data collection and research sample has been enriched and, at the same time, the description of screening tools (AUDIT, GN-SBQ, CAST) has been added. This allowed a better understanding of the measured scores. The Results section has been reorganized in terms of presenting the results. Last but not least, possible explanations for the findings have been provided in the Discussion section. The Discussion section also contains a comparison of pre-pandemic results and pandemic results.
- Response: Your comments and recommendations have really improved the quality of the study. Thank you.
Reviewer 2 Report
The authors present a well-argued manuscript, highlighting the need to study these variables in a particularly vulnerable group and during a period of time in which behaviors related to their mental health have been considerably altered.
Authors should consider the following aspects to improve their manuscript:
The objectives must appear at the end of the introduction section, before methodology.
Since the data collection period coincides with the period of confinement by COVID-19, it would be pertinent for the authors to make a more exhaustive comparison of their results with previous studies that have used the same measurement instruments to highlight whether said confinement has had some effect.
Author Response
Dear Reviewer,
First of all, we would like to thank you for the time spent evaluating the manuscript. We appreciate your evaluation and we consider it to be a significant motivation for our further work. We have carefully considered your comments and incorporated them into the revised manuscript hoping that the changes are in line with your expectations. The responses to your comments and recommendations can be found in the following text and all changes in the revised manuscript are highlighted in yellow.
Your review has been inspiring, and we see that this has greatly improved the quality of our study. We would like to thank you for the constructive feedback and for the opportunity to revise our manuscript.
Kind regards,
Beata Gavurova
Comments and Suggestions for Authors
The authors present a well-argued manuscript, highlighting the need to study these variables in a particularly vulnerable group and during a period of time in which behaviors related to their mental health have been considerably altered.
- Response: Thank you for the summary of the study and for the positive evaluation of the study from the overall point of view. Your view on this issue is inspiring and we really appreciate it. This is our driving force for further research. Thank you for your willingness to help us through your valuable recommendations.
Authors should consider the following aspects to improve their manuscript:
The objectives must appear at the end of the introduction section, before methodology.
- Response: Thank you. Based on your comment, the main objective has been moved from the Methodology section to the end of the Introduction section.
Since the data collection period coincides with the period of confinement by COVID-19, it would be pertinent for the authors to make a more exhaustive comparison of their results with previous studies that have used the same measurement instruments to highlight whether said confinement has had some effect.
- Response: Thank you for your recommendation. The study as a whole has been greatly improved in terms of capturing the COVID-19 period. We have tried to present COVID-19 as a period that needs to be examined and in which the presented research took place. Based on your recommendation, the current version of the Discussion section provides a comparison of the results between the pre-pandemic period and the pandemic period. This step made it possible to capture COVID-19 in terms of time and not its impact on society. At this point, it can be stated that, as this problem is neglected in the examined countries, the pre-pandemic results were limited. Therefore, this study fills a large gap in research in these countries.
Round 2
Reviewer 1 Report
I think the authors made a satisfactory major revision.